# Racial & ethnic disparities in geographic access to critical care in the United States: A geographic information systems analysis

**Kendall J. Burdick**[1]*, **Chris A. Rees**[2], **Lois K. Lee**[3,4], **Michael C. Monuteaux**[3,4], **Rebekah Mannix**[3,4], **David Mills**[3,4], **Michael P. Hirsh**[1], **Eric W. Fleegler**[3,4]

1 Department of Pediatrics, Boston Children's Hospital, Boston, MA, United States of America, 2 Division of Emergency Medicine, Emory University, Atlanta, GA, United States of America, 3 Division of Emergency Medicine, Boston Children's Hospital, Boston, MA, United States of America, 4 Departments of Pediatrics and Emergency Medicine, Harvard Medical School, Boston, MA, United States of America

* kendall.burdick@umassmed.edu

**Data Availability Statement:** The data that support the findings of this study are available from the American Trauma Society and were used under license for the current study. Data are available

## Abstract

### Objective

It is important to identify gaps in access and reduce health outcome disparities, understanding access to intensive care unit (ICU) beds, especially by race and ethnicity, is crucial. Our objective was to evaluate the race and ethnicity-specific 60-minute drive time accessibility of ICU beds in the United States (US).

### Design

We conducted a cross-sectional study using road network analysis to determine the number of ICU beds within a 60-minute drive time, and calculated adult intensive care bed ratios per 100,000 adults. **We** evaluated the US population at the Census block group level and stratified our analysis by race and ethnicity and by urbanicity. We classified block groups into four access levels: no access (0 adult intensive care beds/100,000 adults), below average access (>0–19.5), average access (19.6–32.0), and above average access (>32.0). We calculated the proportion of adults in each racial and ethnic group within the four access levels.

### Setting

All 50 US states and the District of Columbia.

### Participants

Adults ≥15 years old.

### Main outcome measures

Adult intensive care beds/100,000 adults and percentage of adults national and state) within four access levels by race and ethnicity.

upon request from the American Trauma Society. The population data used in the study comes from ESRI and is commercially available with a license for ArcGIS. Similar data is readily available from the Census. Our trauma center location data are owned by a third-party organization can be accessed by others through direct request to the Trauma Information Exchange Program (TIEP). Per their site, "Please email the ATS Staff at tiep@amtrauma.org if you are interested in more detailed TIEP data.

**Funding:** The author(s) received no specific funding for this work.

**Competing interests:** The authors have declared that no competing interests exist.

## Results

High variability existed in access to ICU beds by state, and substantial disparities by race and ethnicity. 1.8% (n = 5,038,797) of Americans had no access to an ICU bed, and 26.8% (n = 73,095,752) had below average access, within a 60-minute drive time. Racial and ethnic analysis showed high rates of disparities (no access/below average access): American Indians/Alaskan Native 12.6%/28.5%, Asian 0.7%/23.1%, Black or African American 0.6%/ 16.5%, Hispanic or Latino 1.4%/23.0%, Native Hawaiian and other Pacific Islander 5.2%/ 35.0%, and White 2.1%/29.0%. A higher percentage of rural block groups had no (5.2%) or below average access (41.2%), compared to urban block groups (0.2% no access, 26.8% below average access).

## Conclusion

ICU bed availability varied substantially by geography, race and ethnicity, and by urbanicity, creating significant disparities in critical care access. The variability in ICU bed access may indicate inequalities in healthcare access overall by limiting resources for the management of critically ill patients.

## Introduction

In a 2010 prescient research letter, Carr, Addyson, and Kahn noted that, "a pandemic or disaster affecting a small proportion of the population could quickly exceed critical care capacity" in the United States (US) [1]. In 2020, this warning was actualized during the COVID-19 pandemic.

Critical care resources should ideally be equitably available to all populations, regardless of their resident state, race, or ethnicity. In 2009, the US had 34.7 adult intensive care unit (ICU) beds per 100,000 adults; however, this ratio was not uniformly distributed throughout the country [2]. Specifically, 93% of ICU beds were located in metropolitan areas, 6% in micropolitan areas, and only 1% located in rural areas of the US [3]. A 2020 study showed gradients of ICU bed availability by income level for people 50 years and older, but did not assess racial and ethnic differences in ICU access in the US nor across the entire adult population [4].

Our objective was to evaluate the racial and ethnic-specific accessibility of ICU beds in the US. We determined the availability of ICU beds per population served within a 60-minute drive time, and then analyzed geographic access based on race and ethnicity and urbanicity. Though access to healthcare is multifactorial, including insurance coverage and socioeconomic means, at a minimum, the ability to reach the facility within a reasonable travel time is essential. Thus, we focused our analysis on geographic access to evaluate who can readily reach ICU level care. The identification of gaps in ICU bed access by state, as well as by race and ethnicity, is essential to target improvements in critical care access and to reduce disparities in healthcare outcomes.

## Methods

### Study design

We conducted a cross-sectional study to determine the ratio of ICU beds to adult population served within a 60-minute drive time at the Census block group level in the US. This study was deemed exempt from review by the institutional review board at Boston Children's Hospital as no identifiable patient data were involved.

## Data sources

We used 2018–2019 American Trauma Society Trauma Information Exchange Program (TIEP), which provides hospital locations and bed counts for each facility [5, 6]. We considered a hospital with at least one adult ICU bed to be an ICU location. Bed counts included total adult ICU beds (sum of medical, surgical, and cardiac) for each location.

We used the 2021 population demographic estimates at the level of the census block group, provided through ArcGIS (ESRI, Redlands, CA). ArcGIS demographic data are based on US Census data and the American Community Survey, and were the most accurate population numbers across all geographies in the US, especially at the census tract and block group geography levels, compared to four other major data vendors [7]. We considered the adult population to be individuals 15 years and older as classified by the American College of Surgeons [8]. Demographic data included the 2021 population by race and ethnicity. Race and ethnic groups included: American Indian or Alaska Native (AI/AN here forward), Asian, Black or African American (Black here forward), Hispanic or Latino (Hispanic here forward), Native Hawaiian and other Pacific Islander (Pacific Islander here forward), and White. The dataset did not delineate race by ethnicity (i.e., "White" was available, but "non-Hispanic White" and "Hispanic White" were not available). Persons who identify as "Two or more races" or as "Some Other Race" were excluded because of the lack of clarity as to whom these designations represent.

## Geographic ICU bed access analysis

**ICU service areas.** We used 60-minute drive time to examine ICU access. Though there is no published standard for appropriate travel time to an ICU, we determined a 60-minute drive time based on the "golden hour" standard for trauma patient transport to reduce mortality, as has been done previously [9–14]. As a sensitivity analysis we conducted separate 30- and 90-minute drive time evaluations.

For each ICU location, we used road network analysis to generate a 60-minute service area, which is a polygon displaying the geographic area served by an ICU within a 60-minute drive. Road network analysis, which has been used in similar studies, [9] is an optimized geographic routing analysis that calculates drive time based on multiple travel-related factors, such as distance, speed limits, road capacity, and stoplights [15]. Therefore, from any point within the generated ICU service area, an adult is able to reach an ICU within a 60-minute drive.

**Block group ICU bed to adult ratio.** We determined the adult population that was served by each ICU at the Census block group level. An ICU was considered accessible to a block group if the block group centroid, the center of an irregular polygon, was within the ICU catchment area. Then, for every block group, we calculated the ratio of adult ICU beds available to the adult population served (see S1 Fig in S1 File for an example). We excluded block groups with populations of 0 in this analysis, since there was no population requiring ICU access.

**Block group access levels.** After assigning an ICU bed ratio to all block groups, we calculated the mean and standard deviation (SD) of the block group bed ratio. We excluded the top 1% of block group ratios in the calculation of the mean, as those typically represented skewed block groups located adjacent to major medical centers. We defined the average ICU bed access as ±0.5 SD of the mean. We then categorized block groups into the following four access levels: (A) no ICU bed access (block group centroid located outside of any 60-minute ICU catchment area), (B) below average access (ICU bed to adults served ratio below 0.5 SD of the mean), (C) average access (ICU bed to adults served ratio within ±0.5 SD of the mean), and (D) above average access (ICU bed to adults served ratio above 0.5 SD of the mean).

## Outcome measure

Our primary outcome measure was the accessibility of ICU beds per population served at the national and state level. For each of the four access levels, we analyzed population racial and ethnic demographics at the Census block group level, then aggregated to represent national and state levels. As urban location likely influences access to ICU bed, we also conducted analyses by urbanicity, based on block group urbanization group (Principle Urban Centers, Urban Periphery, Metro Cities, Suburban Periphery, Semirural, Rural) [7].

## Statistical methods

ArcGIS (ESRI 2021. ArcGIS Pro: Version 2.7. Redlands, CA: Environmental Systems Research Institute) was used to geocode ICU locations, perform road network analysis, and calculate ICU bed access ratios. We then calculated descriptive statistics of ICU access for total, state, and each racial and ethnic group. Excel (version 16.47) was used to create heatmaps to visually compare access by race and ethnicity across states. Since the data used for this analysis comprised the entire US adult population (i.e., it is not a sample drawn from a larger population), inferential statistical evaluations were not warranted.

## Results

There were 3,114 ICU locations, 70,736 adult ICU beds and 273,213,121 adults in the US in 2018. The mean block group-level bed ratio was 25.8 ICU beds per 100,000 adults (SD ± 12.4). For the 2021 population, most adults (71.4%) had average or above average ICU bed access within a 60-minute drive time (Table 1). Overall, increased access generally mirrored higher population density (Figs 1 and 2). The ICU bed ratio in western US states demonstrated low and scattered access.

### No or below average ICU bed access within 60-minute drive time

Overall, 5,038,797 adults (1.8%) had no ICU bed access within a 60-minute drive time (Table 1). For no ICU bed access, AI/AN adults had the highest percentage (12.6%, n = 321,170) and Black adults had the lowest percentage (0.6%, n = 198,477). There were

**Table 1. Population ICU bed access for access levels by race and ethnicity.**

| Race/Ethnicity* | Total** | ICU Beds / 100,000 Adults* | | | | | | | |
|---|---|---|---|---|---|---|---|---|---|
| | | No Access | | Below Average | | Average | | Above Average | |
| | | 0 | | >0–19.5 | | 19.6–32.0 | | >32.0 | |
| | *N* | *n* | *%* | *n* | *%* | *n* | *%* | *n* | *%* |
| Total | 273,213,121 | 5,038,797 | 1.8% | 73,095,752 | 26.8% | 111,551,394 | 40.8% | 83,527,178 | 30.6% |
| AI/AN | 2,557,951 | 321,170 | 12.6% | 729,540 | 28.5% | 849,815 | 33.2% | 657,426 | 25.7% |
| Asian | 16,430,033 | 120,747 | 0.7% | 3,792,732 | 23.1% | 8,603,441 | 52.4% | 3,913,113 | 23.8% |
| Black | 34,282,392 | 198,477 | 0.6% | 5,670,717 | 16.5% | 13,739,382 | 40.1% | 14,673,816 | 42.8% |
| Hispanic | 46,262,773 | 655,254 | 1.4% | 10,636,518 | 23.0% | 22,115,302 | 47.8% | 12,855,699 | 27.8% |
| Pacific Islander | 518,017 | 27,176 | 5.2% | 181,319 | 35.0% | 232,475 | 44.9% | 77,047 | 14.9% |
| White | 194,539,646 | 3,995,993 | 2.1% | 56,345,774 | 29.0% | 76,572,788 | 39.4% | 57,625,091 | 29.6% |

*All racial groups include Hispanic and non-Hispanic ethnicities (ex. White includes Hispanic White and non-Hispanic White persons). Persons who identify as "Two or more races" or as "Some Other Race" were excluded. Population and percentages do not sum to 100% due to people who identify as multiple races and because we could not account for non-Hispanic subgroups.

**US adult population (15 years and older) within 60-minute drive.

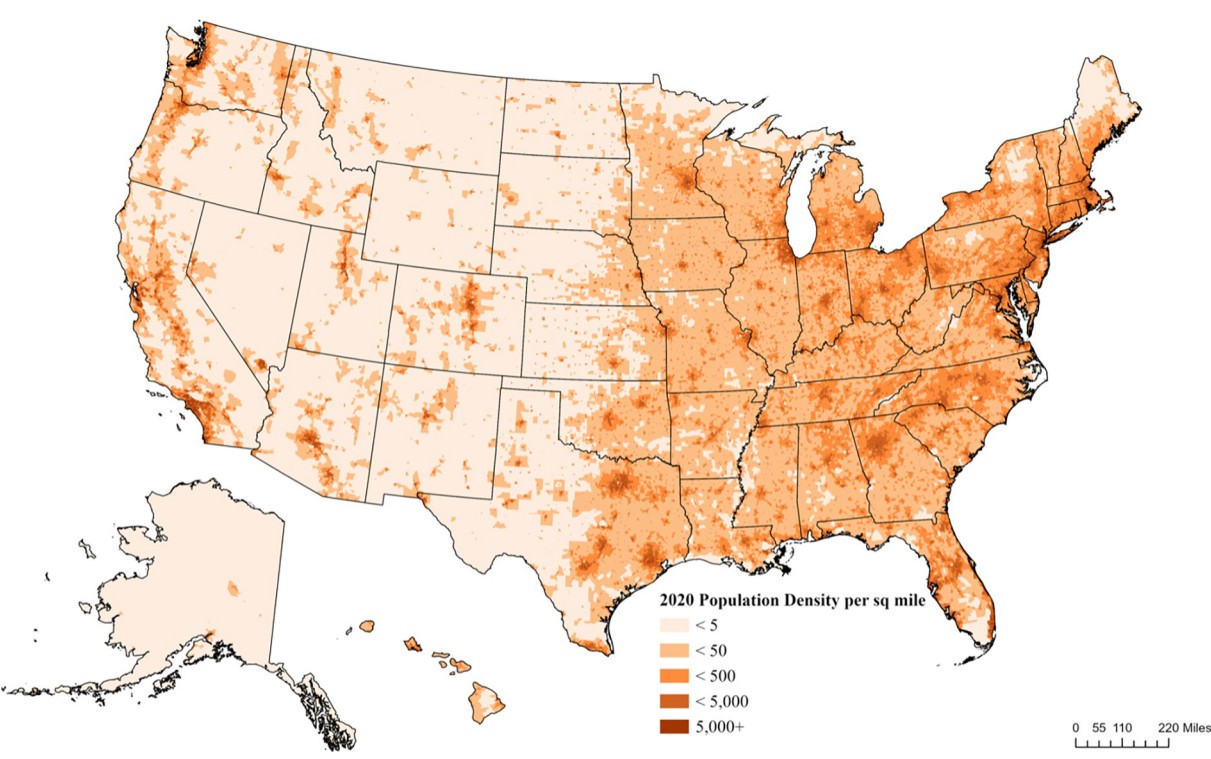

**Fig 1.** 2020 Population density (population per square mile).

73,095,752 adults (26.8%) with below average ICU bed access. Pacific Islander adults had the highest percentage (35.0%, n = 181,319) of below average ICU bed access, and Black adults had the lowest percentage (16.5%, n = 5,670,717).

### Average and above average ICU bed access within 60-minute drive time

There were 111,551,394 adults (40.8%) who had average ICU bed access and 83,527,178 adults (30.6%) who had above average ICU bed access. For average access, AI/AN adults had the lowest percentage (33.2%, n = 849,815) and Asian adults had the highest percentage of average access (52.4%, n = 8,603,441). For above average access, Pacific Islander adults had the lowest percentage (14.9%, n = 77,047) and Black adults had the highest percentage (42.8%, n = 14,673,816) (Table 1).

### State-level disparities in ICU bed access

There were state-level disparities in ICU bed access overall and by race and ethnicity (Fig 3, S2A, S2B Figs in S1 File). When considering the adult population overall, states had a high degree of variability (median [IQR]; minimum–maximum) of no access (1.4% [0.5%– 5.0%]; 0% - 36.9%) and below average (23.5% [15.4%– 36.2%]; 2.1%– 90.9%) ICU bed access. Average ICU bed access (38.9% [26.9%– 54.2%]; 4.9%– 79.0%) across states ranged from a low in Wisconsin at 4.9% to a high in D.C. at 79.0%. Above average ICU bed access (23.3% [6.0%– 45.5%]; 0.0%– 72.3%) was highly variable as well (Fig 3).

Within states, disparities existed by race and ethnicity (S3A and S3B Figs in S1 File). In 13 states, the rate of no access for AI/AN adults was more than double the state's overall rate of no access. In Arizona, AI/AN adults had more than seven-fold higher rates of no ICU access

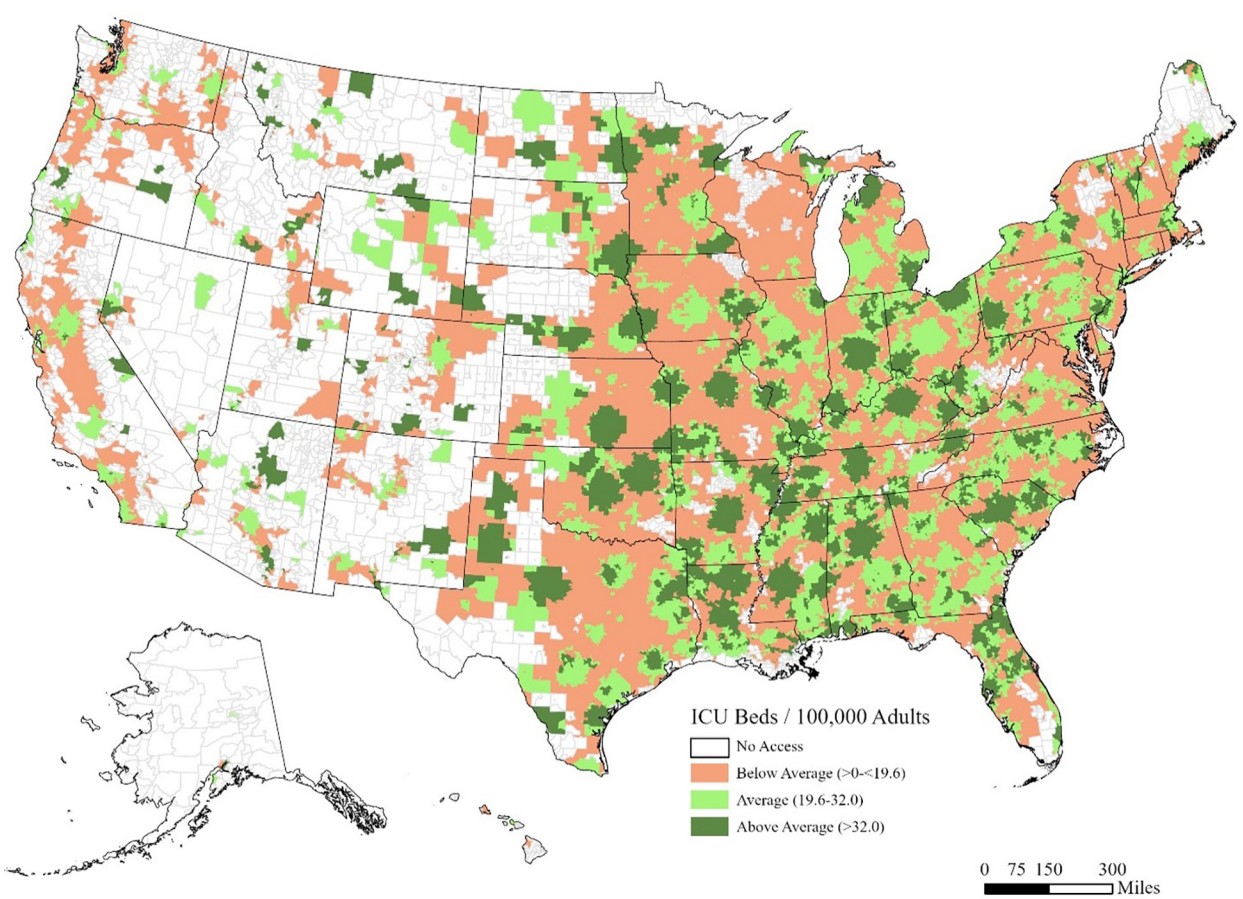

**Fig 2. ICU beds per 100,000 adults within 60-minute drive time at the US Census block group level.** Availability measured as the number of adult ICU beds per 100,000 adults (>15 years old).

than the state's overall adult population (37.6% vs. 5.1%, respectively). AI/AN adults in North Dakota and South Dakota also had high rates of no access compared to the state's overall adult population (53.3% vs. 17.4% and 51.8% vs. 10.5%, respectively). For all racial and ethnic groups, adults in Wisconsin had the highest rates of below average ICU access (AI/AN 90.6%, Asian 96.7%, Black 98.2%, Hispanic 94.1%, Pacific Islander 87.7%, and White 89.9%). Across almost all states, Asian adults had the highest rates of average or above average access.

## Urbanicity

Most block groups had access to an ICU within 60-minute ground access (Fig 2) with lower access in western states. More adults in rural and semirural block groups had no access to an ICU bed (5.2%), compared to more urban block groups (i.e., principle urban centers (0.2%) and urban periphery ((0.7%), Table 2).

## Sensitivity analysis

Findings for our 30-minute and 90-minute sensitivity analyses reflected the similar geographic pattern and racial and ethnic differences. Analyses are shown in S1A and S1B Tables and S3A and S3 Figs in S1 File.

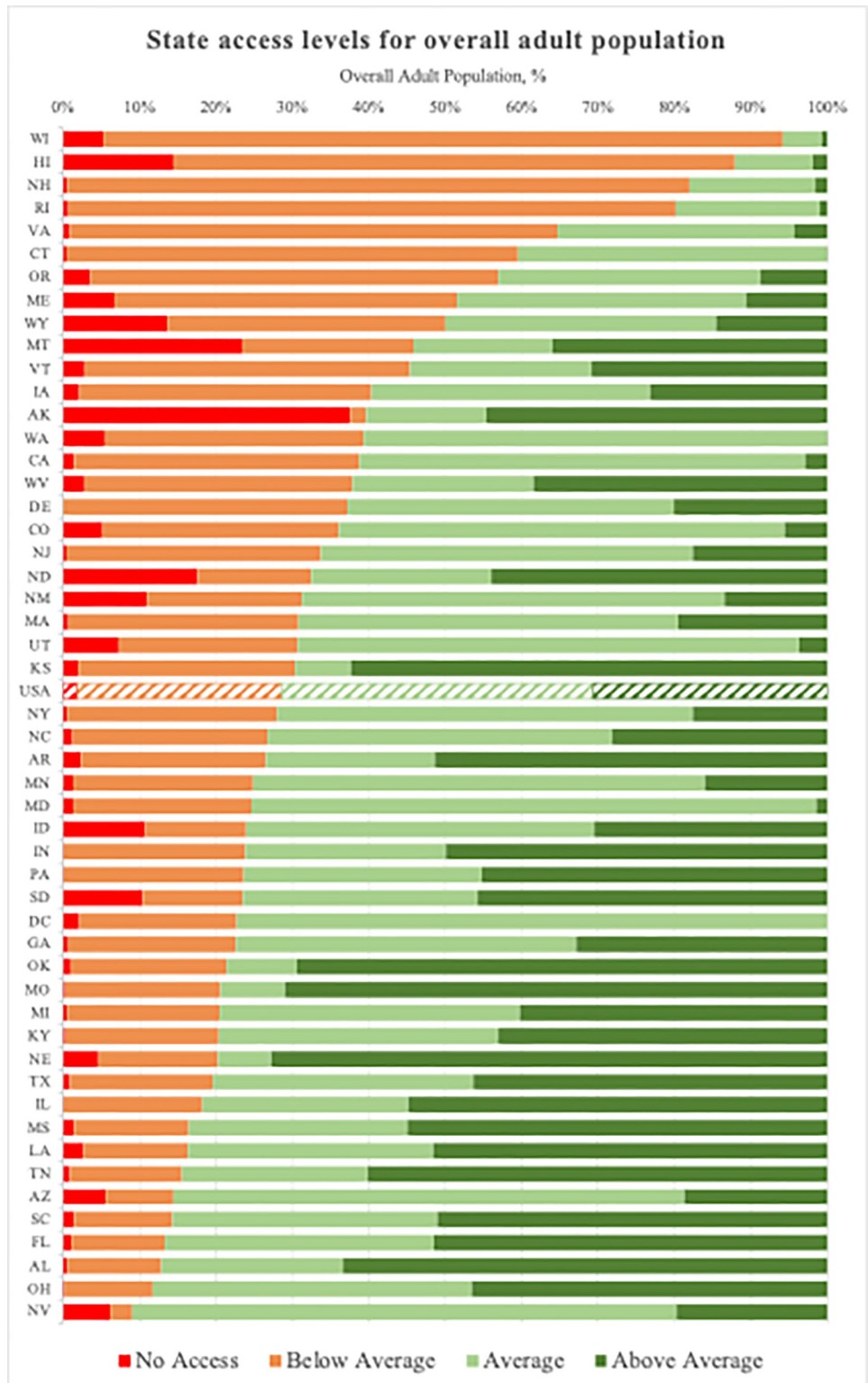

**Fig 3. State access levels for overall adult population.** Availability measured as the number of adult ICU beds per 100,000 adults (>15 years old).

**Table 2. Population ICU bed access by block group urbanization.**

| Urbanization Group | Total adult population | | Population in block group with No Access | | Population in block group with Below Average | | Population in block group with Average | | Population in block group with Above Average | |
|---|---|---|---|---|---|---|---|---|---|---|
| | N (%) | | n (%) | | n (%) | | n (%) | | n (%) | |
| **Total** | 273,213,121 | | 5,038,797 | 1.8% | 73,095,752 | 26.8% | 111,551,394 | 40.8% | 96720985 | 35.4% |
| Principle Urban Centers | 17,999,658 | 6.6% | 34,424 | 0.2% | 2,054,474 | 11.4% | 10,692,960 | 59.4% | 5,217,800 | 29.0% |
| Urban Periphery | 48,286,246 | 17.7% | 327,465 | 0.7% | 8,759,019 | 18.1% | 22,855,529 | 47.3% | 16,344,233 | 33.8% |
| Metro Cities | 45,323,815 | 16.6% | 413,402 | 0.9% | 10,626,712 | 23.4% | 16,964,404 | 37.4% | 17,319,297 | 38.2% |
| Suburban Periphery | 103,116,473 | 37.7% | 974,191 | 0.9% | 24,229,599 | 23.5% | 38,953,952 | 37.8% | 38,958,731 | 37.8% |
| Semirural | 24,900,378 | 9.1% | 839,346 | 3.4% | 8,180,983 | 32.9% | 8,367,029 | 33.6% | 7,513,020 | 30.2% |
| Rural | 46,242,302 | 16.9% | 2,426,233 | 5.2% | 19,053,063 | 41.2% | 13,490,093 | 29.2% | 11,272,913 | 24.4% |
| No Data | 538,056 | 0.2% | 23,736 | 4.4% | 191,902 | 35.7% | 227,427 | 42.3% | 94,991 | 17.7% |

## Discussion

There was a high degree of variability in ICU bed access at the state level as well as large disparities by race and ethnicity, as well as by urbanicity, across the US. While the majority of US adults had average or above average ICU bed access, over 5 million US adults had no access and over 73 million had below average access to an ICU bed within a 60-minute drive time. AI/AN and Pacific Islander adults had substantially lower ICU bed access, compared to individuals of other racial and ethnic groups. Large access differences existed across states with less access in the rural block groups and western US.

There are limitations to our analysis. First, our geographic analysis of access is limited to drive time. There is no published and recognized standard for ICU transport time, so our 60-minute analysis was extrapolated based on the established "golden hour" trauma transport time and expanded by two sensitivity analyses which demonstrate similar disparities in access at 30- and 90-minute drive times. Second, we did not use air transport in this analysis, which is more common in the western US and Alaska. The goal of the analysis was to understand the density of ICU availability at the population level. Air transport for ICU patients is typically between facilities, rather than from patient homes and surroundings; and therefore, was excluded from this analysis. Finally, our results make no causal claims relating ICU bed access to patient outcomes, which are due to multiple factors including the disease or injury itself, co-morbidities, quality of care and other resources. Despite these limitations, our results provide new insights into the accessibility of ICU beds and adult populations with, and without, ready access to ICU beds in the US.

The implications for limited ICU bed access are life-threatening, as seen during the COVID-19 pandemic [16]. The disparities in access to ICU beds by race and ethnicity may play a contributing role in the racial and ethnic health inequities in critical conditions. For example, our study demonstrated AI/AN and Pacific Islander adults had limited ICU bed access, while both populations have been found to have a greater risk of morbidity and mortality from COVID-19 [17, 18]. Furthermore, racial minorities have a disproportionately higher mortality rate from sepsis and acute respiratory failure, both of which commonly (if not exclusively) require ICU care [19, 20]. While these mortality differences are multifactorial and are associated with insurance status, [21] neighborhood conditions, [22] poverty, [23, 24] access to healthcare, [25] and underlying health conditions, [26] the availability of resources such as ICU beds is essential for prompt and adequate treatment.

Travel time to a facility is not the only metric to evaluate healthcare access. For example, individuals may live within an appropriate drive time to a facility, but still not receive the same

quality of care if they are uninsured or underinsured. Since there were 30.8 million US citizens who were uninsured in May 2020, [27] lack of insurance coverage is a major barrier to healthcare access. Though Black adults had higher geographic access to ICU beds, prior studies suggest that Black adults are twice as likely to be uninsured, compared to White adults [28]. The causes of disparities in healthcare access and outcomes also include key elements of structural racism such as high levels of individual and community poverty that disproportionality effect racial and ethnic minorities [26, 29]. While these factors contribute to the quality of care and the ability to receive care, geographic access to a facility with resources proportional to the population served is a minimum, and necessary, requirement.

In 2020, Kanter *et al*. showed gradients of ICU bed availability by income level for people 50 years and older, based on 3,160 hospital service areas, and found an average of 49.7 ICU beds per 100,000 individuals over the age of 50 [4]. They found that low-income communities have fewer ICU beds per 100,000 compared to wealthier communities, with this difference more pronounced in rural areas. They argued for greater policy support and coordination of care during the COVID-19 pandemic. Our study adds to their findings by analyzing the entire US adult population and stratifying by race and ethnicity.

No consensus currently exists on the standard number of ICU beds needed or on the geographic catchment area of an ICU's served population. Variability in ICU bed access exists throughout the world and the US [1, 2, 30]. In 2004, the US, Canada and the United Kingdom had 20.0, 13.5 and 3.5 ICU beds per 100,000 adults, respectively.[30] Most countries have a similar ratio of hospital to ICU beds, per capita; however, the US deviates from this trend–heavily favoring ICU beds over hospital beds. ICU beds in the US have increased since 1985, but hospital capacity overall remained constant due to a concurrent decline in non-ICU hospital beds [31].

Having too few ICU beds can result in denial or delay of ICU admission, premature discharges, longer hospital stays, and higher mortality rates [32–35]. In contrast, limiting ICU beds may decrease hospital costs for critical care, which currently represents 17.4%-39.0% of all hospital costs [31]. Studies suggest that a high number of ICU beds may have a deleterious effect, as increased ICU capacity is not directly associated with improved health outcomes and may cause nosocomial risk to patients [36–38]. A ceiling effect may exist after which increased ICU capacity may divert hospital resources away from valuable lower acuity beds [39]. However, it is also likely that a floor effect is present where a critical threshold must be met in terms of adequate ICU bed availability. There are also challenges in establishing more ICU beds in rural and critical access areas with fewer resources and less reliable use of these beds, for the financial justification of these beds.

Prior studies have shown disparities in access to key resources such as COVID-19 vaccination sites, pediatric trauma centers, dialysis units and cardiac rehabilitation centers [13, 40–42]. It is clear from this data that critical care resources are also not equitably available to all areas and populations of the US. Specifically, the western US, and some racial and ethnic groups, such as AI/AN and Pacific Islander adults, have disproportionately lower than average access, which may result in poorer outcomes and long-term health problems. There is an important need for the extension of resources to provide equitable critical care resources for a broadly distributed population across all states and racial and ethnic groups.

## Conclusions

Current ICU bed availability varies substantially nationally and by state, creating significant disparities in access to critical care resources. AI/AN and Pacific Islander populations had considerably less access to ICU beds in the US. These disparities may contribute to worse health

outcomes in certain regions and populations in the US. This analysis is only one perspective on access; however, it is an important step to highlight the inequalities of healthcare access in the US. These data emphasize a need for enhanced critical care resources in multiple states in order to equitably serve the entire US population. Only by analyzing and addressing the physical and policy barriers to healthcare access can we work towards reducing healthcare disparities by region and by racial and ethnic groups in the US.

## Supporting information

**S1 File. Contains supporting figures and tables.**
(DOCX)

## Acknowledgments

We thank Jeff Blossom, at the Center for Geographic Analysis at Harvard University, Boston, for mapping support.

## Author Contributions

**Conceptualization:** Kendall J. Burdick, Chris A. Rees, Lois K. Lee, Michael C. Monuteaux, Rebekah Mannix, David Mills, Michael P. Hirsh, Eric W. Fleegler.

**Data curation:** Kendall J. Burdick, Eric W. Fleegler.

**Formal analysis:** Kendall J. Burdick, Michael C. Monuteaux.

**Investigation:** Kendall J. Burdick, Eric W. Fleegler.

**Methodology:** Kendall J. Burdick, Chris A. Rees, Lois K. Lee, Michael C. Monuteaux, Rebekah Mannix, David Mills, Michael P. Hirsh, Eric W. Fleegler.

**Project administration:** Michael P. Hirsh, Eric W. Fleegler.

**Resources:** Eric W. Fleegler.

**Software:** Kendall J. Burdick.

**Supervision:** Eric W. Fleegler.

**Validation:** Kendall J. Burdick, Michael C. Monuteaux, Eric W. Fleegler.

**Visualization:** Kendall J. Burdick, Eric W. Fleegler.

**Writing – original draft:** Kendall J. Burdick, Eric W. Fleegler.

**Writing – review & editing:** Kendall J. Burdick, Chris A. Rees, Lois K. Lee, Michael C. Monuteaux, Rebekah Mannix, David Mills, Michael P. Hirsh, Eric W. Fleegler.

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
