## [Decision Letter · Decision Letter 0]

15 Mar 2023

PONE-D-23-00264Racial & Ethnic Disparities in Geographic Access to Critical Care in the United States: A Geographic Information Systems AnalysisPLOS ONE

Dear Dr. Burdick,

Thank you for submitting your manuscript to PLOS ONE. After careful consideration, we feel that it has merit but does not fully meet PLOS ONE’s publication criteria as it currently stands. Therefore, we invite you to submit a revised version of the manuscript that addresses the points raised during the review process.

We look forward to receiving your revised manuscript.

Kind regards,

Jean Baptiste Lascarrou

Academic Editor

PLOS ONE

Journal Requirements:

2. You indicated that ethical approval was not necessary for your study. We understand that the framework for ethical oversight requirements for studies of this type may differ depending on the setting and we would appreciate some further clarification regarding your research. Could you please provide further details on why your study is exempt from the need for approval and confirmation from your institutional review board or research ethics committee (e.g., in the form of a letter or email correspondence) that ethics review was not necessary for this study? Please include a copy of the correspondence as an ""Other"" file.

4. We note that Figures 1, 2, S1 and S2 in your submission contain map images which may be copyrighted. All PLOS content is published under the Creative Commons Attribution License (CC BY 4.0), which means that the manuscript, images, and Supporting Information files will be freely available online, and any third party is permitted to access, download, copy, distribute, and use these materials in any way, even commercially, with proper attribution. For these reasons, we cannot publish previously copyrighted maps or satellite images created using proprietary data, such as Google software (Google Maps, Street View, and Earth). For more information, see our copyright guidelines: http://journals.plos.org/plosone/s/licenses-and-copyright.

a. You may seek permission from the original copyright holder of Figures 1, 2, S1 and S2 to publish the content specifically under the CC BY 4.0 license.  

Additional Editor Comments:

None

Reviewers' comments:

Reviewer's Responses to Questions

**Comments to the Author**

1. Is the manuscript technically sound, and do the data support the conclusions?

Reviewer #1: Partly

Reviewer #2: Yes

2. Has the statistical analysis been performed appropriately and rigorously? 

Reviewer #1: Yes

Reviewer #2: Yes

3. Have the authors made all data underlying the findings in their manuscript fully available?

Reviewer #1: Yes

Reviewer #2: No

4. Is the manuscript presented in an intelligible fashion and written in standard English?

Reviewer #1: Yes

Reviewer #2: Yes

5. Review Comments to the Author

Reviewer #1: The investigators present an analysis that aims to evaluate racial and ethnic disparities in access to intensive care services in the United States. The main analytic approach evaluates accessibility, using four categories of ICU bed counts per capita within a 60-minute drive from the centroid of each census block. They then calculated the proportion of adults in each racial and ethnic category within each access category.

1. The overall conclusion is that there is ICU bed availability according to geography, race, ethnicity, and urbanicity. This is not a novel finding and it diminishes my enthusiasm for the paper.

2. The numbers of intensive care unit beds, hospital beds, and hospitals are changing to a significant extent each year, so a cross sectional analysis in a single year is a less informative measure of access.

3. Not all intensive care unit beds are the same. Access to the services of a critical access hospital is not the same as access to a major medical center. A more nuanced evaluation of access to intensive care units that have full services would have been more informative.

4. Though others have used the same drive-time analysis described in this paper, an improvement that restricted driving to within-state would have more closely mirrored the behavior of EMS agencies.

5. As public health policy and Medicaid services are typically carried out at the state level, an analysis restricted to the state may have identified disparities resulting from policy.

Reviewer #2: Kendall J. Burdick J. K. et al, write an interesting paper about racial and Ethnic Disparities in access to ICU beds. The article is well written and provides insights for decision policy.

MINOR comments : Where does demographic data come from? Please could you specify in your article. What does ESRI demographic data mean? In the demographic data, Are Ethnicity and race self-reported?

Which type of ICU beds were included in the analysis? Can you provide the definition of ICU beds?

Maybe it would be interesting to stratified by the number of hospital? It's quite different in terms of disparities and equalities of access of care, if you compare for exemple 100 ICU beds of 1or2 hospitals (high volume hospital with high number of ICU beds) than 100 ICU beds of 10 hospitals (smaller hospitals but more widespread in the country).

6. PLOS authors have the option to publish the peer review history of their article (what does this mean?). If published, this will include your full peer review and any attached files.

Reviewer #1: No

Reviewer #2: **Yes: **Richard CHOCRON

---

## [Author Response · Author response to Decision Letter 0]

31 Mar 2023

Thank you for your thorough review of our manuscript and thoughtful comments. We have reviewed your comments, made the appropriate changes to our manuscript, and responded to your comments below. Thank you again for your consideration of our manuscript for publication in PLOS One.

We have updated our formatting to match the requested formatting document. 

2. You indicated that ethical approval was not necessary for your study. We understand that the framework for ethical oversight requirements for studies of this type may differ depending on the setting and we would appreciate some further clarification regarding your research. Could you please provide further details on why your study is exempt from the need for approval and confirmation from your institutional review board or research ethics committee (e.g., in the form of a letter or email correspondence) that ethics review was not necessary for this study? Please include a copy of the correspondence as an ""Other"" file.

Since our study did not include patient data, and the Census data through ArcGIS is publicly available, our study was deemed exempt by our IRB office. Please see the submitted IRB letter for confirmation. 

The population data used in the study comes from ESRI and is commercially available with a license for ArcGIS. Similar data is readily available from the Census. Our trauma center location data are owned by a third-party organization can be accessed by others through direct request to the Trauma Information Exchange Program (TIEP). Per their site, “Please email the ATS Staff at tiep@amtrauma.org if you are interested in more detailed TIEP data.”

4. We note that Figures 1, 2, S1 and S2 in your submission contain map images which may be copyrighted. All PLOS content is published under the Creative Commons Attribution License (CC BY 4.0), which means that the manuscript, images, and Supporting Information files will be freely available online, and any third party is permitted to access, download, copy, distribute, and use these materials in any way, even commercially, with proper attribution. For these reasons, we cannot publish previously copyrighted maps or satellite images created using proprietary data, such as Google software (Google Maps, Street View, and Earth). For more information, see our copyright guidelines: http://journals.plos.org/plosone/s/licenses-and-copyright.

Figures 1, 2, S1 and S2 are original and were created using the ArcGIS Pro software by author, Kendall Burdick. There is no copyright to these images. 

As above, we have updated our file citations and added the supporting information to the end of our manuscript.

Additional Editor Comments:

None

Reviewers' comments:

Reviewer's Responses to Questions

Comments to the Author

1. Is the manuscript technically sound, and do the data support the conclusions?

Reviewer #1: Partly

Reviewer #2: Yes

2. Has the statistical analysis been performed appropriately and rigorously? 

Reviewer #1: Yes

Reviewer #2: Yes

3. Have the authors made all data underlying the findings in their manuscript fully available?

Reviewer #1: Yes

Reviewer #2: No

4. Is the manuscript presented in an intelligible fashion and written in standard English?

Reviewer #1: Yes

Reviewer #2: Yes

5. Review Comments to the Author

Reviewer #1: The investigators present an analysis that aims to evaluate racial and ethnic disparities in access to intensive care services in the United States. The main analytic approach evaluates accessibility, using four categories of ICU bed counts per capita within a 60-minute drive from the centroid of each census block. They then calculated the proportion of adults in each racial and ethnic category within each access category.

1. The overall conclusion is that there is ICU bed availability according to geography, race, ethnicity, and urbanicity. This is not a novel finding and it diminishes my enthusiasm for the paper.

Thank you for the time spent reviewing and assessing our manuscript. While disparities are well described throughout many sectors of medicine, a national analysis of ICU access, at the population level, stratified by race, ethnicity and urbanicity has not be described in the literature before. This analysis provides detailed, population-specific analysis and a map of the US where critical care services can be provided, relative to their population.

2. The numbers of intensive care unit beds, hospital beds, and hospitals are changing to a significant extent each year, so a cross sectional analysis in a single year is a less informative measure of access.

It would be desirable to evaluate these data over time instead of only a cross section analysis; however, this historic and annual data was not available to us at the time of this analysis. But we still feel that the disparities which our analysis emphases are an important contribution that will help identify not only where ICU beds are needed but also how the distribution of these beds affects people of different races and ethnicities.

3. Not all intensive care unit beds are the same. Access to the services of a critical access hospital is not the same as access to a major medical center. A more nuanced evaluation of access to intensive care units that have full services would have been more informative.

There is variation in ICU bed type, for example, medical, surgical, and cardiac. In this study, we condensed the 3 types of beds into one category for our calculations. We believe that this is an appropriate methodological approach, considering that, as you mention in your previous comment, ICU beds have the ability to adapt into different types (ie surgical transforming into medical beds during the COVID surges), and this study is an assessment of critical care availability as a whole. 

4. Though others have used the same drive-time analysis described in this paper, an improvement that restricted driving to within-state would have more closely mirrored the behavior of EMS agencies.

We agree that the restriction to within-state analysis would mirror EMS agency support; though we did not limit our analysis to this, as care can be provided our reached through self-transport. Additionally, in the state of surges and hospital overflow, geographic analysis provides a comprehensive assessment of care regardless of state lines. It has been well documented that patients are frequently moved across state lines for ICU level care.

5. As public health policy and Medicaid services are typically carried out at the state level, an analysis restricted to the state may have identified disparities resulting from policy.

We agree that the state level analysis provides interesting and unique insights into access, which is why we provided the state level variation in our supplemental material. 

Reviewer #2: Kendall J. Burdick J. K. et al, write an interesting paper about racial and Ethnic Disparities in access to ICU beds. The article is well written and provides insights for decision policy.

MINOR comments : Where does demographic data come from? Please could you specify in your article. What does ESRI demographic data mean? In the demographic data, Are Ethnicity and race self-reported?

Our data came from ArcGIS, which is provided by the company ESRI. This data is based on US Census data and the American Community Survey, and has been validated as the most accurate population data for US geographical research. We have clarified our method language to explain this.

Methods: We used the 2021 population demographic estimates at the level of the census block group, provided through ArcGIS (ESRI, Redlands, CA). ArcGIS demographic data are based on US Census data and the American Community Survey, and were the most accurate population numbers across all geographies in the US, especially at the census tract and block group geography levels, compared to four other major data vendors.

Which type of ICU beds were included in the analysis? Can you provide the definition of ICU beds?

As mentioned in the Methods, ICU beds were described by the American Trauma Society Trauma Information Exchange Program database and included adult ICU beds, which was the sum of medical, surgical, and cardiac ICU beds.

Methods: We considered a hospital with at least one adult ICU bed to be an ICU location. Bed counts included total adult ICU beds (sum of medical, surgical, and cardiac) for each location. 

Maybe it would be interesting to stratified by the number of hospital? It's quite different in terms of disparities and equalities of access of care, if you compare for example 100 ICU beds of 1or2 hospitals (high volume hospital with high number of ICU beds) than 100 ICU beds of 10 hospitals (smaller hospitals but more widespread in the country).

We agree that this would add an interesting detail to analysis. We will be sure to include this stratification in future research.

---

## [Editor Report · Decision Letter 1]

12 Jun 2023

Racial & Ethnic Disparities in Geographic Access to Critical Care in the United States: A Geographic Information Systems Analysis

PONE-D-23-00264R1

Dear Dr. Burdick,

We’re pleased to inform you that your manuscript has been judged scientifically suitable for publication and will be formally accepted for publication once it meets all outstanding technical requirements.

Kind regards,

Jean Baptiste Lascarrou

Academic Editor

PLOS ONE

Additional Editor Comments (optional):

- None

Reviewers' comments:

- None

---

## [Editor Report · Acceptance letter]

26 Jun 2023

PONE-D-23-00264R1 

Racial & Ethnic Disparities in Geographic Access to Critical Care in the United States: A Geographic Information Systems Analysis 

Dear Dr. Burdick:

I'm pleased to inform you that your manuscript has been deemed suitable for publication in PLOS ONE. Congratulations! Your manuscript is now with our production department. 

Kind regards, 

on behalf of

Dr. Jean Baptiste Lascarrou 

Academic Editor

PLOS ONE